# Microstructure Evolution and Mechanical Stability of Supersaturated Solid Solution Co-Rich Nanocrystalline Co-Cu Produced by Pulsed Electrodeposition

**DOI:** 10.3390/ma13112616

**Published:** 2020-06-08

**Authors:** Killang Pratama, Jenifer Barrirero, Frank Mücklich, Christian Motz

**Affiliations:** 1Chair of Materials Science and Methods, Saarland University, 66123 Saarbrücken, Germany; motz@matsci.uni-sb.de; 2Department of Metallurgical Engineering, Institute Technology of Bandung, Bandung 40132, Indonesia; 3Chair of Functional Materials, Saarland University, 66123 Saarbrücken, Germany; j.barrirero@mx.uni-saarland.de (J.B.); muecke@matsci.uni-sb.de (F.M.)

**Keywords:** nanocrystalline Co-Cu, spinodal decomposition, micro mechanics, mechanical stability

## Abstract

Thick films of supersaturated solid solution nanocrystalline Co-Cu (28 at.% Cu) were synthesized through the pulsed electrodeposition technique. Microstructural changes of nanocrystalline Co-Cu were intensively studied at various annealing temperatures. Annealing at 300 °C results in a spinodal decomposition within the individual grains, with no grain coarsening. On the other hand, distinct phase separation of Co-Cu is detected at annealing temperatures beyond 400 °C. Static micro-bending tests show that the nanocrystalline Co-Cu alloy exhibits a very high yield strength and ductile behavior, with no crack formation. Static micro-bending tests also reported that a large plastic deformation is observed, but no microstructure change is detected. On the other hand, observation on the fatigue resistance of nanocrystalline Co-Cu shows that grain coarsening is observed after conducting the cyclic micro-bending test.

## 1. Introduction

Studies on the fundamental behavior [1], mechanical properties [2], and possible industrial applications [3] of nanocrystalline materials are wide-spread in the literature. Among of the promising materials in this class are supersaturated solid solution nanocrystalline alloys, e.g., Co-Cu alloy, the microstructure and phase distribution of which can be modified by annealing treatments, to enhance thermal and mechanical stability, as well as tailoring physical properties [4,5]. Investigations into the thermal stability and microstructure evolution of nanocrystalline Co-Cu have been performed in previous works, mostly at temperatures beyond 400 °C [6,7,8], but more research about microstructural changes at lower temperatures is required. The information about the mechanical properties of nanocrystalline Co-Cu is also limited [9,10,11]. On the other hand, the information about the mechanical properties of single-phase nanocrystalline Co and Cu is widely available in the literature [12,13,14,15,16,17]. Therefore, more research on the mechanical properties of nanocrystalline Co-Cu with different microstructures is required.

Nanocrystalline pure Co [12,13] and Cu [14,15,16,17] possess excellent mechanical properties compared to the conventional coarse-grained counterpart. However, slight reductions in ductility were observed, due to flaws induced during synthesis and surface preparation, as well as low strain hardening capability and low resistance to crack initiation and propagation [18,19,20]. In addition, mechanical instability during fatigue test of nanocrystalline and ultrafine grained materials was also observed and intensively studied in the literature [21,22,23]. Considering the mechanical instability of nanocrystalline materials, strategies were developed for improvement, such as the utilization of non-uniform grain sizes [20,24] and secondary phase or particles [25,26]. Among the promising strategies is the application of multiple phases in the nanocrystalline materials [25,26], which are also termed metallic composites, in which different dislocation activity or deformation mechanisms in the primary and secondary phases are expected to improve the plastic deformation behavior of nanocrystalline materials.

The formation of metallic composite and precipitation of solute components from immiscible alloys including the Co-Cu system is possible through the spinodal decomposition [27,28]. However, spinodal decomposition and nucleation in nanocrystalline Co-Cu at high temperature (>400 °C) is usually followed by grain coarsening, such as demonstrated in previous research [6,7]. Thus, investigation of the spinodal decomposition to modify the microstructure at low temperature is needed, to minimize the grain coarsening effect. The spinodal decomposition of Co and Cu in nanocrystalline Co-Cu is crucial to produce multiple phase materials. It has been reported that nanocrystalline Cu has a good ductility, while nanocrystalline Co has a significant high strength [12,13,14,15,16,17]. Interestingly, cyclic mechanical loading shows that nanocrystalline Cu [23] and Co [29] have a different softening-hardening behavior. The different mechanical behavior of nanocrystalline Co and Cu can be used as an interesting combination for improving the mechanical stability and performance of nanocrystalline Co-Cu system.

In this research work, nanocrystalline Co-Cu were produced through the pulsed electrodeposition (PED) technique, instead of severe plastic deformation (SPD). According to the literature [30,31,32,33], lattice defects in SPD- and PED-processed nanocrystalline materials (e.g., excess vacancies, dislocation, and twins) are different in characteristic, type, and number, thus, different mechanical properties are expected. Furthermore, to investigate the impacts of the synthesis route on the microstructural changes upon heat treatment, comprehensive studies on the microstructure evolution of electrodeposited nanocrystalline Co-Cu annealed at various temperatures were performed in this study, by scanning electron microscope (SEM), transmission electron microscope (TEM), X-ray diffraction (XRD), and atom probe tomography (APT). The principal mechanical properties are investigated by microhardness measurements. At the end of the paper, the mechanical stability of nanocrystalline Co-Cu and the effect of plastic deformation on the microstructure are studied, through static and cyclic micro-bending tests.

## 2. Materials and Methods

Nanocrystalline Co-Cu films up to a thickness of 400 µm were deposited through PED on Cu disk cathodes (surface diameter: 12 mm) through a Pt anode under specific parameters and conditions of deposition: current density: 1000 A/m^2^; pulse period: 2 ms; break period: 18 ms; temperature: 40 °C. The deposition was performed in an electrolyte containing 112.20 g/L CoSO_4_.7H_2_O, 6.38 g/L CuSO_4_, 56.44 g/L C_4_H_4_KNaO_6_.4H_2_O, 142.04 g/L Na_2_SO_4_, 18.55 g/L H_3_BO_3_, 2.00 g/L C_7_H_5_NO_3_S (Saccharin), and 0.20 g/L C_12_H_25_NaO_4_S (sodium dodecyl sulfate). In the present work, the parameters of deposition and composition of electrolyte were based on previous work by Müller [34], and were optimized to obtain compact and homogeneous deposits with a Cu content of 28-at.%.

The deposited samples were subjected to isothermal annealing in vacuum at a pressure of 10^−6^ mbar at various temperatures (300–600 °C) for different periods of time (1–64 h) to investigate the thermal stability and structural transformation. Microstructure and composition analysis were taken from the cross-sections of the samples and carried out in the field-emission scanning electron microscope (SEM) (Zeiss Sigma-VP, Jena, Germany) equipped with backscattered electrons (BSE) and energy dispersive spectroscopy (EDS) detectors at acceleration voltages of 17–20 kV. Grain size determination was performed manually by measuring the length of line interception within individual grains at two different orientations with the imageJ software (Ver 1, LOCI, University of Wisconsin, Madison, USA) [35], and EDS analysis were conducted with the AZtec software (Oxford Instrument Inc., Abingdon, England). Microhardness measurements (HV 0.2; test force: 1.962 N) were performed at cross-sections of the samples with a DuraScan hardness testing device (Struers, Copenhagen, Denmark) and equipped with ecos Workflow software (EMCO-TEST Prüfmaschinen GmbH, Kuchl, Austria). Hardness measurements were performed at positions of 30 µm and 200 µm from the substrate. X-ray diffraction (XRD) measurements were conducted to investigate the phase composition of the individual samples. Cu K-alpha radiation (λ: 1.5405980 Å) and a scan step size of 0.013°2θ/s was used. Transmission electron microscope (TEM) samples were prepared from the polished samples (diameter: 3 mm) in planar mode, and then further processed by GATAN PIPS ion milling to produce an electron transparent film at the center of the disc. TEM observation was performed in a JEOL JEM 2011 (JEOL, Tokyo, Japan) at an accelerating voltage of 200 kV to acquire enhanced microstructural information, as well as diffraction patterns of selected samples.

Specimens for atom probe tomography (APT) were prepared in a dual-beam focused ion beam/scanning electron microscopy workstation (FIB/SEM) (Helios NanoLab 600™, FEI Company, Hillsboro, OR, USA) by the lift-out technique described in the literature [36]. Material was extracted from the center of the cross section of the film. An electron beam induced Pt-capping layer was first deposited to provide protection from gallium implantation. After lift-out and thinning of the specimens, a low energy milling at 2 kV was performed to minimize Ga induced damage [36]. Laser Pulsed APT was carried out in a LEAP 3000X HR (CAMECA, Gennevilliers, France). Measurements were performed at repetition rates of 100 and 160 kHz, a pressure lower than 1.33 × 10^−8^ Pa, an evaporation rate of 5 atoms per 1000 pulses and a specimen temperature of about 60 K. The laser used has a wavelength of 532 nm, a pulse length of 10 ps, and a pulse energy of 0.5 nJ. Datasets were reconstructed and analyzed with IVAS™3.6.8 software (CAMECA, Gennevilliers, France). Regions-of interest (ROI) were used to construct one-dimensional concentration profiles through the specimens. Three APT specimens of the as-deposited alloy and four of the annealed sample were analyzed.

Micro bending beams were manufactured in a focused ion beam (FIB/SEM) (Helios NanoLab 600™, FEI Company, Hillsboro, OR, USA) on the cross-section of the nanocrystalline film at a position of 200 µm from the substrate. The dimensions of the beams are presented in Table 1. One micro bending beam (beam-A) was prepared for a static micro bending test, while cyclic micro bending tests were performed on beam-B and beam-C at different plastic strain amplitudes. In situ static and cyclic micro-bending test were performed in a TESCAN-Vega scanning electron microscope (TESCAN, Brno, Czech Republic) equipped with the Advanced Surface Mechanics (ASMEC, Ulm, Germany) nanoindentation system and InspectorX Ver. 2 UNAT software (ASMEC, Ulm, Germany) to record the force-displacement data. A nano-indenter and a double blade griper were used to impose static and cyclic mechanical loading onto micro bending beams at the test parameters, which are given in Table 1. The force-displacement data were calculated and presented as a surface stress (σ_s_) and a surface strain (ε_s_), based on elastic bending beam theory through Equations (1) and (2). The recorded force and displacement are F and D, respectively, while L_B_, W, and T correspond to length, width, and thickness of the bending beam, respectively.
(1)$$\sigma_{s}=\frac{{6L}_{B}F\;}{{W\;T}^{2}}$$
(2)$$\varepsilon_{s}=\frac{D\;T}{{2L}_{B}^{2}}$$

## 3. Results and Discussion

### 3.1. Structural Evolution

Figure 1a depicts a cross section image and EDS map of compact and porosity-free deposit of nanocrystalline Co-Cu film deposited at a current density of 1000 A/m^2^ and a pulse period 2 ms (duty cycle: 10%). The EDS line profile in Figure 1b shows a quiet homogeneous concentration of Co and Cu along a cross-section of nanocrystalline Co-Cu (28 at.% Cu) film. The EDS line profile confirms that the cobalt concentration drops up to ~15 at.% from area near the Cu-substrate to the surface of nanocrystalline film. Thus, deposition for more than 400 µm of thicknesses is not preferred, due to the inhomogeneous concentration of Co and Cu. Deposition of nanocrystalline Co-Cu at a higher Cu concentration is possible by reducing current density. However, a back-scattered electron image of Co-Cu (45 at.% Cu) alloy, deposited at a current density of 500 A/m^2^ and a pulse period of 2 ms (duty cycle: 10%), shows an inhomogeneous microstructure. in which a mixture of ultrafine grained and nanocrystalline structure are observed (see Figure 1c). Deposition at a higher current density and duty cycle [for example at current density of 1500 A/m^2^ and duty cycle 20% (see Figure 1d)], lead to the formation of microporous (mark with blue arrows in see Figure 1d) and an inhomogeneous chemical composition. Therefore, to achieve homogeneous microstructure and chemical composition, as well as compact and porosity-free deposits, all samples used in this paper were deposited at a current density of 1000 A/m^2^ and pulse period of 2 ms (duty cycle: 10%).

Figure 2a shows a combined bright-field TEM image and selected area diffraction (SAD) pattern of the PED-processed nanocrystalline Co-Cu (28 at.% Cu), in which nanometer sized grains of about 22.7 ± 8.2 nm are observed (Figure 2b). A number of twins are also observed, which are marked with red arrows in Figure 2a. The formation of *twins* is believed to be due to effect of chemical additives (saccharine and sodium dodecyl sulfate) during the electrodeposition process, as demonstrated in the previous paper [33]. The contribution from high current density and pulsed current may also have an influence on the formation twins. The XRD (Figure 2c) pattern shows that the ED nanocrystalline Co-Cu (28 at.% Cu) exhibits a supersaturated solid solution phase, in which the peaks of {111} and {200} planes of the single fcc phase are observed. The as-deposited nanocrystalline Co-Cu (28 at.% Cu) exhibit a high hardness of 4.45 ± 0.07 GPa. The high value of microhardness could be contributed to by several factors, such as chemical composition, grain size, and number of lattice defects (twins, dislocations, and vacancies). The co-deposition of impurity elements (e.g., carbon, sulfur, and hydrogen) also has an influence on the hardness of the electrodeposited nanocrystalline Co-Cu.

In the present work, early stages of microstructural changes of the PED-processed nanocrystalline Co-Cu are studied at 300 °C, which minimizes grain coarsening, as shown in previous research [6,7]. All samples annealed at 300 °C for different periods (1 h, 5 h, and 24 h) exhibit an increase of hardness, compared to the as-deposited state (see Table 2). Interestingly, no significant grain coarsening is observed in the sample annealed even for 24 h in which nano sized grains of less than 50 nm remain unchanged (see Figure 2d). It is believed that the hardening effect is influenced by the spinodal decomposition of the solid solution Co-Cu, as demonstrated by Kato [37]. The PED-processed nanocrystalline Co-Cu was further annealed at 300 °C for up to 64 h, to study its long-term thermal stability and the chemical decomposition. Figure 2e depicts a combined bright-field TEM image and SAD pattern of an at 300 °C for 64 h annealed sample, showing no significant grain changes nor grain coarsening. Figure 2e also shows the presence of twins (indicated by red arrows). In this annealed sample, microstructure and hardness observations show that the average grain size is about 27.2 ± 7.9 nm (see Figure 2f) and the hardness increases up to 4.83 ± 0.07 GPa. In comparison with the as-deposited state, only minor changes are detected from the XRD measurement (Figure 2c). Comparing the SAD pattern of as deposited (Figure 2a) and annealed (Figure 2e) samples, lattice shifts of the {111} and {200} planes are observed, and the spinodal decomposition of the Co-Cu solid solution at the nanometer scale during annealing at 300 °C is expected to cause the shift. Consequently, peak shifts of the {111} and {200} planes are observed from the XRD pattern (see Figure 2c). However, it is also believed that the lattice shift is not single-handedly caused by spinodal decomposition, but could also be caused by other factors or processes during isothermal annealing at 300 °C for 64 h, such as the relaxation of internal stresses and grain boundaries, reducing of defect density, etc. However, an investigation on internal stress, grain boundaries relaxation, and defect density in nanocrystalline Co-Cu is not conducted in this paper, in which the observation is focused on the spinodal decomposition of nanocrystalline Co-Cu. Thus, APT measurements are conducted to investigate the possible structural and chemical decomposition at 300 °C.

APT measurements were conducted for the as deposited and annealed (300 °C for 64 h) samples. Figure 3a shows an APT elemental map of a slice through a reconstruction of the as-deposited sample, and a concentration profile corresponding to the white dashed line. According to these results, the as-deposited sample shows a solid solution with imperfect homogeneous concentration of Co and Cu at nanometer scale, but no short-range fluctuation (e.g., spinodal decomposition, precipitation) of Co or Cu is found. This was also observed previously on the high-pressure torsion (HPT)-processed nanocrystalline Co-Cu [7,38], and it was also shown that spinodal decomposition with minor grain coarsening was observed at 400 °C [7]. In the present work, APT measurements show that a spinodal decomposition of Co-Cu takes place also at 300 °C. Figure 3b depicts an APT elemental map of a selected slice from a reconstruction of an annealed sample. Three regions with distinctive compositions are observed: (i) Co-Cu solid solution, (ii) Co-rich regions (up to 95 at.% Co) and Cu-rich region (up to 90 at.% Cu). Moreover, an APT elemental map of another reconstruction in the annealed sample (Figure 3c) shows small compositional fluctuations in the solid solution region. Such compositional fluctuations are crucial characteristic of a spinodal decomposition [39], and these fluctuations will grow continuously until a metastable equilibrium is achieved.

The atomic diffusion at the grain boundaries is not considered here, since the compositional fluctuations occurs within the grains at 300 °C. Of course, diffusion at grain boundaries is much faster and can lead to a fast de-mixing in these regions, which is possibly the reason for the “spots” with high Co or Cu concentration in e.g., Figure 3b. However, here, it is focused on the atomic diffusion of Co and Cu, which induces a spinodal decomposition within individual grains. The diffusion rate in this nanocrystalline supersaturated solid solution may be different compared to common materials, as the defect content, lattice constant, crystal structure, etc. may be different. However, a rough estimation of the atomic mobility of the alloys’ components can be considered for an initial study. In previous research [7], atomic mobility of solute atom Co in the nanocrystalline Co-Cu (74 at.% Cu) was calculated through the classical theory of diffusion equation $L=\sqrt{\mathrm{Dt}}$, where L, D, and t are described as distance, diffusion coefficient, and period of diffusion (time), respectively. In the present work, the atomic mobility of Co and Cu atoms in a Co-Cu solid solution system (28 at.% Cu) is also roughly estimated through the same procedure. Considering the chemical composition of the alloy system (72 at.% Co and 28 at.% Cu), the possible diffusion mechanisms that are likely to occur are Co self-diffusion, Cu diffusion in Co, and interdiffusion of Co-Cu. The diffusion coefficient data were taken from the literature [40,41,42]. However, since the diffusion coefficient at 300 °C is not available, the diffusion coefficient data from the literature [40,41,42] were extrapolated through the Arrhenius equation (D = A exp [−E_A_/RT]) to get the diffusion coefficient data at 300 °C (it is assumed that the diffusion mechanism doesn’t change). The calculated diffusion distance for 300 °C and 64 h from the mechanisms of Co self-diffusion, Cu diffusion in Co, and Co-Cu interdiffusion are 0.0005 nm, 0.0014 nm, and 0.098 nm, respectively. The measurements show that the atomic mobility of Co and Cu is very low, in which diffusion distances of less than 1 nm were calculated. Surprisingly, the APT measurement (Figure 3b) shows that the diffusion distance of Co and Cu atoms is in the range of 5 to 10 nm. At this point, it must be considered that all diffusion coefficients [40,41,42] were measured in bulk materials, so the result would be different in nanocrystalline materials, due to the larger number of lattice defects within a grain as mentioned before. The contribution from mainly vacancy-type defects in nanocrystalline materials may significantly enhance the atomic diffusivity [43].

Figure 4a,b show microstructure images of electrodeposited nanocrystalline Co-Cu (28 at.% Cu) annealed at 450 °C for 5 h and 24 h. A significant grain coarsening and hardness decrease are observed from these micrograph (see also Table 2). Interestingly, no significant differences of grain size and hardness values are observed between samples annealed for short (1 h and 5 h) and long (24 h) periods. The XRD pattern of a sample annealed at 450 °C for 5 h shows a shoulder on the peak of {111} planes (marked with black arrow in Figure 5a), but no significant phase separation is detected. It is believed that the observed shoulder is an indication of an early stage of spinodal decomposition where Cu-rich regions is formed. In contrast, XRD pattern of a sample annealed at 450 °C for 24 h exhibits a massive phase separation of solid solution Co-Cu in which fcc-Cu, fcc-Co, and hcp-Co {101} peaks are detected (see Figure 5a). However, since the peaks of fcc-Co and fcc-Cu are hard to separate, it is believed that a certain amount of solid solution Co-Cu phase remained, and that phase separation is in progress. In addition, a number of particles (black spots in the images) are observed mainly at the grain boundary, and twins are also recognized from the micrograph (Figure 4b). Based on some evidences, microstructure and phase evolution of the electrodeposited nanocrystalline Co-Cu (28 at.% Cu) at 450 °C can be divided into different stages. During short annealing periods (1 h and 5 h), the diffusion of Co and Cu atoms leads predominantly to spinodal decomposition and grain boundary movement. The early stage of spinodal decomposition leads to the formation of Co and Cu concentration fluctuations within the grains. At the grain boundaries, where the diffusion is expected to be faster, the formation Cu-precipitates can be expected. It is believed that the dark particles that are observed in the Figure 4a,b are related to these Cu-precipitates. These Cu-precipitates are believed to block the grain boundaries movement (pinning of grain boundaries), thus, a constant grain size is obtained after a certain period of annealing. Upon annealing for longer times (24 h), the spinodal decomposition progresses, and a phase separation will also take place within the grains. The Cu precipitates at the grain boundaries will still hinder their movement, hence the grain size remains almost constant.

Figure 4c,d show the micrograph of electrodeposited nanocrystalline Co-Cu (28 at.% Cu) annealed at 600 °C for 5 h and 24 h. In comparison with the sample annealed at 450 °C, the grain coarsening is more pronounced, thus, as expected from the Hall-Petch relation, the microhardness values slightly decrease to lower than 4 GPa (Table 2). The XRD measurements show that the phase separation is clearly visible in the samples annealed at 600 °C for 5 h and 24 h (Figure 5b), in which the peaks of fcc-Co, fcc-Cu and a weak hcp-Co {101} are detected. According to the microstructure images (Figure 4c,d), the presence of twins and particles (which is believed as Cu-precipitates) are also detected, and their size increases with increasing the annealing period. The phase separation is almost complete, and results in a microstructure consisting of almost pure Co and Cu regions. The mobility of the grain boundaries is rather high at 600 °C and the Cu-precipitates are unable to block the grain boundary move.

According to the XRD and SEM measurements, the microstructural evolution during the annealing of the PED-processed nanocrystalline Co-Cu (28 at.% Cu) is comparable with the HPT-processed nanocrystalline Co-Cu (25 at.% Cu) [6]. Figure 6a,b show the microhardness evolution of PED- and HPT-processed [6,8] nanocrystalline Co-Cu at almost identical chemical composition annealed at different annealing temperatures and times. Upon a short annealing time of 1 h (Figure 6a), the PED-processed nanocrystalline Co-Cu shows an increased hardness at temperatures lower than 450 °C in comparison with the HPT-processed [6] materials. Moreover, Figure 6a,b show that the hardening effect due to chemical decomposition of Co-Cu at 300 °C is more obvious for the PED-processed materials [6,8]. The thermal stability of nanocrystalline materials can be influenced by several factors, which are the initial grain size, phase stability, and stabilization by solute components. The initial grain size of the PED-processed nanocrystalline Co-Cu in the present work is three to four times smaller than the HPT-processed Co-Cu [6,8]. On one hand, this results in an increased driving force for grain growth, however, on the other hand, the small grain size may speed up the segregation of Cu to grain boundaries which may result in early formation of “precipitates” at the boundaries. These small Cu-rich regions may effectively pin the boundaries and avoid grain coarsening which results, along with the spinodal decomposition within the grains, in an increase in hardness at low annealing temperatures. Of course, other mechanisms, like different defect types and densities or the relaxation of boundaries may also contribute to the difference between the PED- and HPT-processed materials. Furthermore, in the PED-processed nanocrystalline materials, additional improvement of thermal stability can also be attributed to the co-deposition of impurity elements, such as carbon and sulfur. In the literature [44], it has been reported that the segregation of sulfur to grain boundaries could significantly improve thermal stability of the PED-processed nanocrystalline Co.

### 3.2. Mechanical Stability

The mechanical properties and microstructure development during plastic deformation of the PED-processed nanocrystalline Co-Cu (28 at.%) were investigated through static and cyclic micro bending experiments. Figure 7a depicts an engineering surface stress vs.- strain curve of micro beam-A resulted from a static micro bending test at a surface strain rate of 1.24 × 10^−2^ s^−1^. Practically, the determination of strain hardening coefficient, total surface strain, and ultimate strength is possible from the surface stress-strain curve. However, it should be carefully considered that the calculated surface stress-strain data through the elastic bending beam theory is only accurate for up to the yield point, and will fail in the plastic regime. Therefore, in the present work, the discussion on the parameters from the surface stress-strain curve will be focused only up to yield point. According to the curve, the measured yield strength is extremely high (σ_y_ ≈ 3.9 GPa). However, it is believed that the actual yield strength is lower, due to the inhomogeneous deformation in bending tests. At the initial stage of deformation, only a small surface layer of the micro beam is plastically deformed (the whole beam is not plastically deformed yet). Thus, the elastic-plastic transition regime is quite smooth, and an exact determination of the yield point is difficult (see Figure 7b). As a consequence, it can be assumed that the yield point is between 2.8 and 3.9 GPa (see Figure 7b). In comparison, according to Tabor’s rule, the yield strength can be estimated from the microhardness measurements with σ_y_ ≈ 1.5 GPa.

Figure 8a,b show the secondary electron (SE) images of micro beam-A after deformation recorded from the top and side view of the bending direction (y-axis). According to Figure 8a,b micro beam-A is plastically deformed, in which no crack formation is detected. Necking effects due to plastic deformation, which are marked with white arrows, are clearly visible (see Figure 8a). The surface changes due to plastic deformation can be easily identified too (see Figure 8b). Therefore, according to the stress-strain curve and SE-detector images, it is obvious that the ED nanocrystalline Co-Cu (28 at.%) exhibits a ductile behavior. Unfortunately, the microstructure observation through the back-scattered electron (BSE) detector is difficult, due to distinct surface roughness. Thus, FIB polishing was performed to produce a smooth surface at the selected areas on micro beam-A. Figure 8c,d depict BSE images of the microstructure at two different positions (zone-A and zone-B) at the FIB polished region. According to Figure 8c,d no microstructural changes, such as grain coarsening, are observed at the areas of large plastic deformation.

In comparison with nanocrystalline Co [12,13], the yield strength of the PED-processed nanocrystalline Co-Cu (28 at.% Cu) is higher. In all pure nanocrystalline materials (including nanocrystalline Co), the strengthening mechanism is dominated by grain boundary strengthening (the Hall-Petch effect). On the other hand, additional strengthening mechanisms are contributing in nanocrystalline Co-Cu (28 at.% Cu) alloys, like solid solution strengthening, including the effect of the spinodal decomposition and precipitation hardening in the case of annealed samples, thus, a higher yield strength is observed. It is assumed that the cobalt content is also an important influencing factor on the improvement of yield strength in nanocrystalline Co-Cu. For example, the PED-processed nanocrystalline Co-Cu (28 at.% Cu) exhibits a higher yield strength, compared to the HPT-processed nanocrystalline Co-Cu (74 at.% Cu) tested by tensile test (σ_y_ ≈ 0.8 GPa) [9], whereas this significant difference can mainly be attributed to the higher Co content in the PED-processed material. Different mechanical test methods also contribute to the significant different values of yield strength. However, it is also believed that the differences may be caused also by the different type and number of lattice defects on between PED- and HPT-processed nanocrystalline Co-Cu.

The number of dislocations in the PED- and HPT-processed nanocrystalline materials is different, due to the dissimilar route of material processing. The formation of dislocations in the HPT-processed materials is predominantly affected by a large plastic strain during the processing of nanocrystalline materials [30]. In comparison, previous reports show that there are two main contributing factors to the formation of dislocations in the PED-processed nanocrystalline materials, which are stress development and relaxation generated by pulsed current [45,46] and grain growth inhibition by chemical additives [33,47]. It has been also reported in the literature [32] that the electrodeposited nanocrystalline Ni-Mo alloy exhibits a slightly different dislocations density than the HPT-processed nanocrystalline Ni-Mo at comparable chemical composition and crystallite size. The influence of chemical additives during the pulse electrodeposition process of nanocrystalline materials does not only contribute to the formation of dislocations, but also to the formation of twins. Figure 2a shows a number of twins in the ED nanocrystalline Co-Cu (28 at.% Cu), which may also contribute to the enhanced strength. The generation of dislocations from twins is expected to enhance the number of dislocations as reported from the literature [48,49]. All in all, the combination of different types of strengthening mechanisms: (i) solid solution, (ii) twins, (iii) dislocations, and (iv) the co-deposition of impurity elements contribute to an enhancement of strength in the ED nanocrystalline Co-Cu (28 at.% Cu).

The mechanical stability during cyclic loading of the ED nanocrystalline Co-Cu (28 at.% Cu) was investigated through the cyclic micro-bending experiment. A cyclic micro-bending test at a low plastic surface strain amplitude ($\varepsilon_{s,a}$ = 2.0 × 10^−4^) was conducted at micro beam-B for 5000 cycles. Figure 9a depicts the surface stress amplitude level ($\sigma_{s,\;\mathrm{amp},\;\mathrm{avg}}$) as a function of number of load cycles of micro beam-B. The measurement shows that no crack formation is found after 5000 load cycles, but gradual softening is clearly visible after 100 load cycles (marked with a red arrow in Figure 9a). Microstructure investigation was conducted at the initial condition and after 5000 load cycles (Figure 9c,d). By comparing the initial and final microstructure, minor grain coarsening is observed. It is strongly believed that the observed grain coarsening causes the softening of the ED nanocrystalline Co-Cu (28 at.% Cu) materials.

Micro beam-C was subjected to four stages of cyclic micro-bending with stepwise increased plastic surface strain amplitudes. According to the results, no crack formation is observed after cycling from stage-I to stage-III, in which the observed surface stress amplitude level ($\sigma_{s,\;\mathrm{amp},\;\mathrm{avg}}$) as a function of number of load cycles remain constant (Figure 9a). Micro beam-C was further subjected to cyclic bending test at stage-IV ($\varepsilon_{s,a}$ = 7.0 × 10^−3^; 100 cycles) and crack formation was observed after the test. Figure 9e depicts a secondary electron image of micro beam-C after testing at stage-IV, which shows that two cracks with different length are formed in two different positions. According to the Figure 9a, at stage-IV, the surface stress amplitude level ($\sigma_{s,\;\mathrm{amp},\;\mathrm{avg}}$) remains constant for up to 40 load cycles, however, the amplitude starts to decline at the 41st of load cycle (marked with black arrow in Figure 9a), and the decrease is more significant at the following load cycles. It is believed that crack formation may start after load cycle 40 at stage-IV, and is followed by crack propagation.

Figure 9b shows the cyclic hysteresis loops of micro beam-C at selected load cycles of stage-IV. It is evident that only a small shift of the surface stress amplitude between the 10th and 55th load cycle is observed in quadrant 1 (Q1). This shift may be attributed to a crack formation, and since the shift of the surface stress amplitude is only in Q1 (Figure 9b), it is supposed that only one crack was formed after 55 load cycles. Afterwards, the surface stress amplitude is significantly altered at the 70th and 90th load cycles in Q1. At the 70th of load cycle, the shift of surface stress amplitude is more significant in Q1, and a small shift is also observable in Q3, which could be an indication that a second crack has formed. At the 90th load cycle, the shifts of the surface stress amplitude are more noticeable in Q1 and Q3, which means that crack has propagated further. After 100 load cycles, it is found that the crack length on the upper and lower side of micro beam-C are ~750 nm and ~1.2 µm, respectively. Figure 9f shows that significant grain coarsening is observed in the region adjacent to the cracks-I, in which some ultrafine grains are visible. It is supposed that the grain coarsening is induced by the local plastic deformation, which is generated during crack propagation. This local plastic deformation could induce grain boundary migration, as also demonstrated on micro beam-B in the present work, and in ultrafine-grained Cu during the cyclic micro bending test [23].

In the present work, it has been demonstrated that structural instabilities during cyclic loading are observed due to grain coarsening. Therefore, improvements are required to enhance the mechanical stability. It has been demonstrated in Section 3.1 that annealing treatment of nanocrystalline Co-Cu could widely modify the microstructure, thus, an improvement in mechanical stability is expected. In the future, research will focus on the influences of different structural characteristics obtained by annealing treatment on the mechanical stability of very fine grain Co-Cu alloy systems.

## 4. Summary

Investigations on microstructure and phase evolution of supersaturated solid solution nanocrystalline Co-Cu (28 at.% Cu), produced through the pulsed electrodeposition technique, have been performed through subsequent annealing treatments at different temperatures for different periods. APT showed that spinodal decomposition of Co-Cu system could start at a low annealing temperature (300 °C), in which grain coarsening is negligible. This chemical decomposition can be utilized to improve mechanical properties of nanocrystalline Co-Cu. Annealing at 450 °C induced significant grain coarsening, which results in ultrafine-grained structure (~125 nm). However, the microhardness could be maintained at comparable level with the as-deposited state, due to the combination of pinning of grain boundaries and phase separation. On the other hand, a gradual decrease of microhardness due to grain coarsening was found after annealing at 600 °C, although a massive phase separation occurred. The obtained various microstructures and phases through the annealing treatment can be utilized to get different mechanical behavior of very fine grain Co-Cu system, hence, the ductility and mechanical stability of very fine grain Co-Cu system could be enhanced.

The mechanical stability of deposited nanocrystalline Co-Cu (28 at.% Cu) was also investigated through static and cyclic micro bending tests. The micro beam specimens showed that the deposited nanocrystalline alloy exhibits ductile properties, in which a very high yield strength was observed. The presence of solid solution strengthening, dislocations, twins, and co-deposition of impurities elements may contribute to the high strength of electrodeposited nanocrystalline Co-Cu. Although this material exhibits very good mechanical properties, the microstructure was not stable in the fatigue tests, and further improvements can be performed here, possibly by optimizing the microstructure by annealing treatments.

## Figures and Tables

**Figure 1 materials-13-02616-f001:**
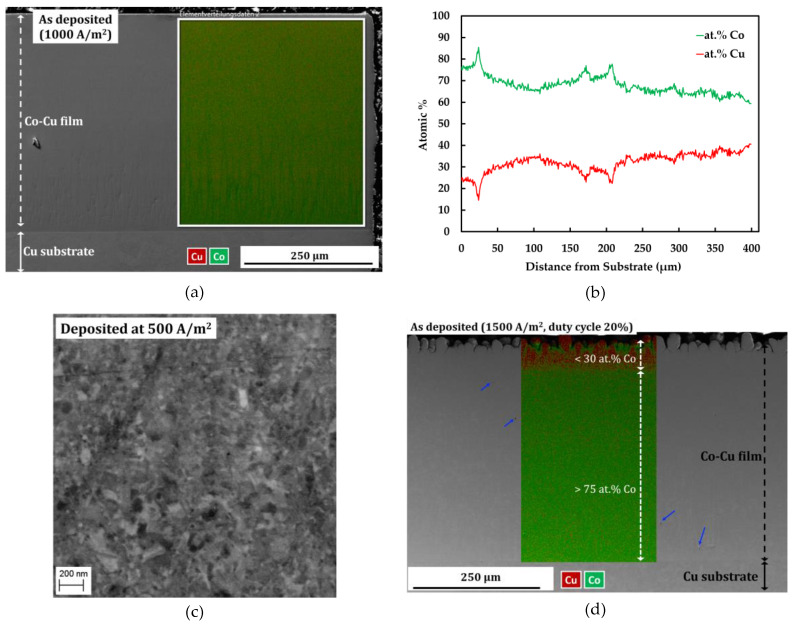
(**a**) Combined secondary electron (SE) image—energy dispersive spectroscopy (EDS) map and (**b**) EDS line profile of sample deposited at a current density of 1000 A/m^2^, and a pulse period of 2 ms (duty cycle: 10%) in the cross-section mode. (**c**) Back-scattered electron images of the microstructure of a sample deposited at a current density of 500 A/m^2^, and a pulse period 2 ms (duty cycle: 10%). (**d**) Combined secondary electron image-EDS map of sample deposited at a current density of 1500 A/m^2^, and a pulse period of 2 ms (duty cycle: 20%) in cross section mode.

**Figure 2 materials-13-02616-f002:**
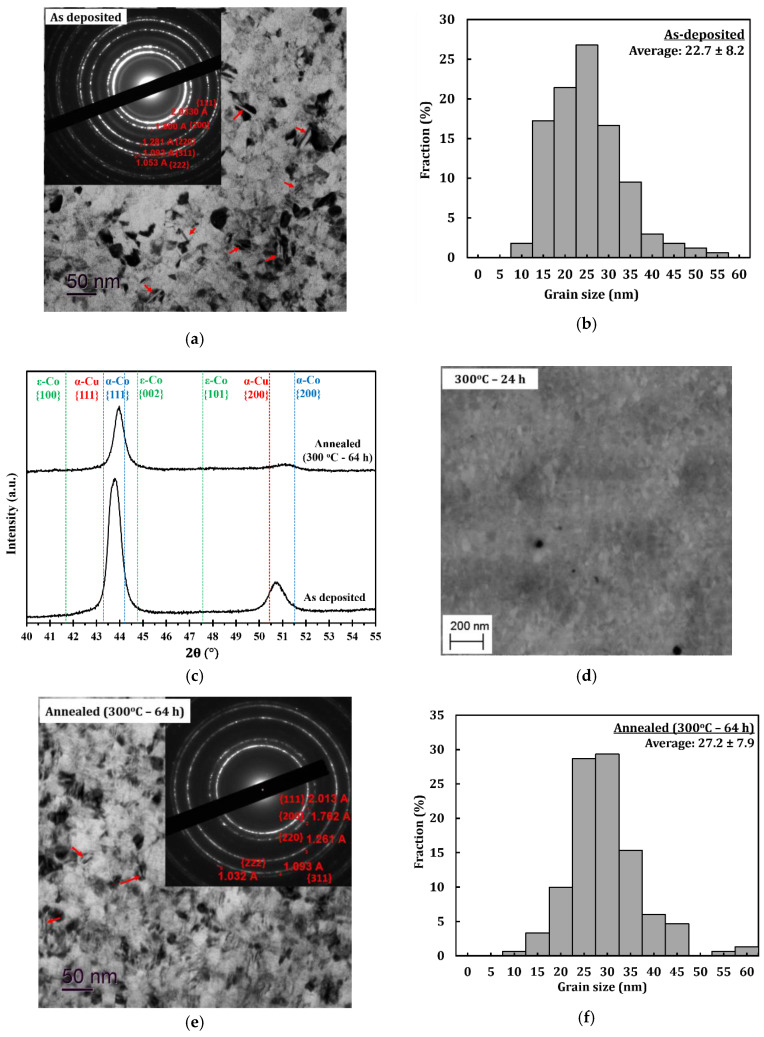
(**a**) Combined bright-field transmission electron microscope (TEM) and selected area diffraction (SAD) pattern images of sample deposited at current density of 1000 A/m^2^ and pulse period 2 ms (duty cycle: 10%) in planar mode and (**b**) its grain size distribution. (**c**) X-ray diffraction (XRD) pattern of as deposited sample and annealed (300 °C–64 h) samples. (**d**) Back-scattered electron images of the microstructure of a sample annealed at 300 °C for 24 h in cross section mode. (**e**) Combined bright-field TEM and selected area diffraction (SAD) pattern images of annealed (300 °C–64 h) sample in planar mode and its (**f**) its grain size distribution.

**Figure 3 materials-13-02616-f003:**
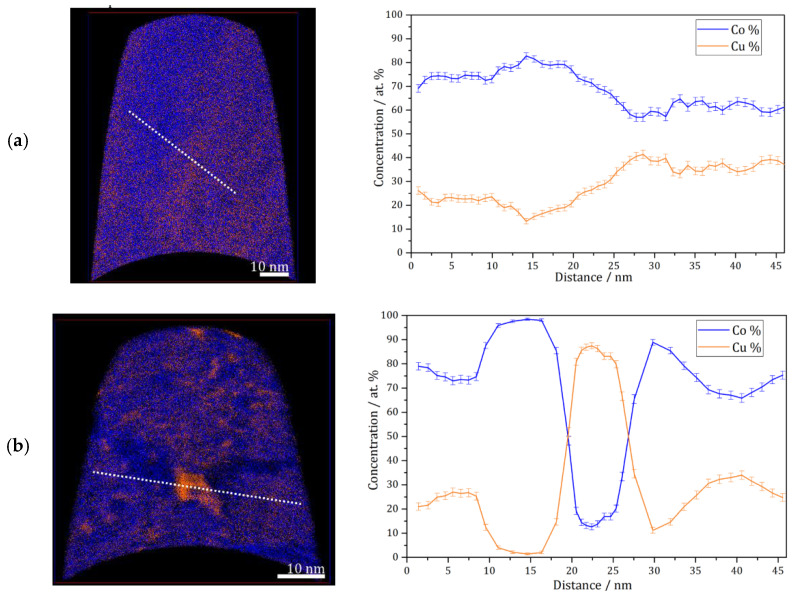
Atom probe tomography (APT) elemental maps of Co (blue) and Cu (orange) of 5 nm thick slices through the reconstructions and one-dimensional concentration profiles along the white dashed lines on the elemental maps (from **left** to **right**). (**a**) As-deposited sample; (**b** and **c**) similar annealed sample (300 °C–64 h).

**Figure 4 materials-13-02616-f004:**
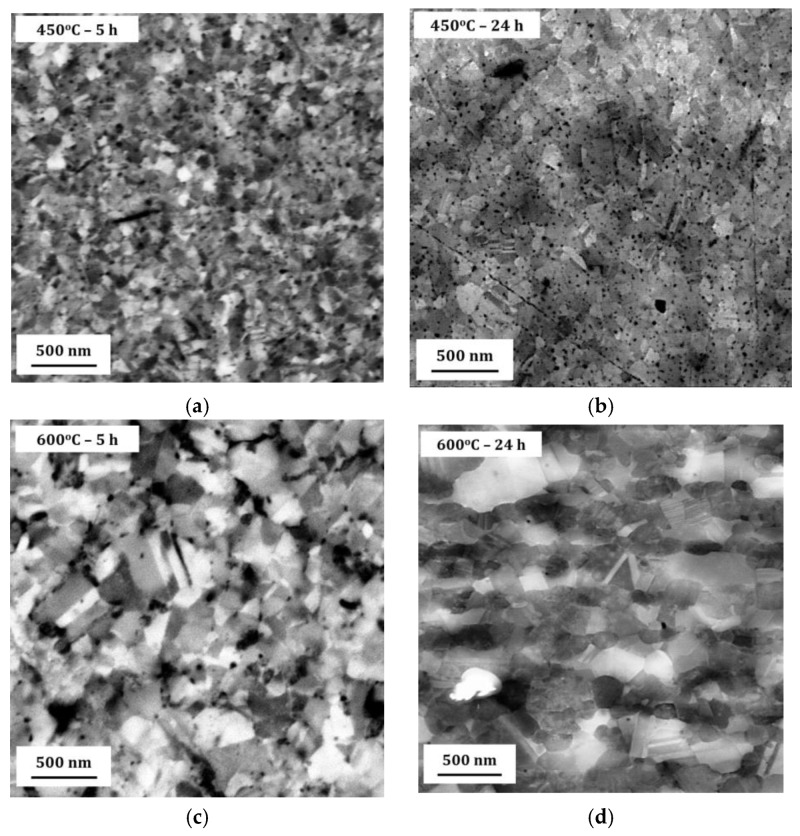
Back-scattered electron images of the microstructure of annealed samples at various temperatures for different periods: (**a**) 450 °C for 5 h, (**b**) 450 °C for 24 h, (**c**) 600 °C for 5 h, (**d**) 600 °C for 24 h. All microstructure images were taken in the cross section of the deposited nanocrystalline Co-Cu films. For individual images, notice the scale bar to distinguish the magnification.

**Figure 5 materials-13-02616-f005:**
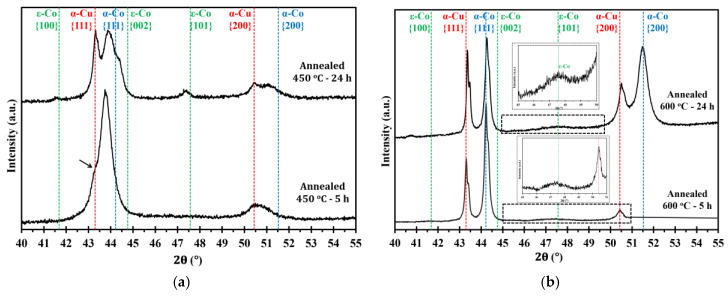
XRD patterns of electrodeposited nanocrystalline Co-Cu (28 at.% Cu) annealed at (**a**) 450 °C and (**b**) 600 °C for short (5 h) and long (24 h) periods of annealing. All measurements were taken in the cross section mode.

**Figure 6 materials-13-02616-f006:**
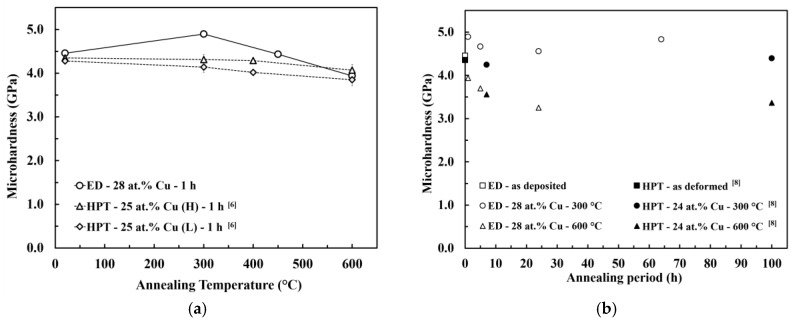
Microhardness evolution of electrodeposited (ED) nanocrystalline Co-Cu (28 at.% Cu) during isothermal annealing at different temperatures for different periods which is compared with HPT-processed nanocrystalline Co-Cu at chemical composition of (**a**) 25 at.% Cu [6] and (**b**) 24 at.% Cu [8]. (H) and (L) are indicating high and low purity of Co and Cu powder used in HPT process.

**Figure 7 materials-13-02616-f007:**
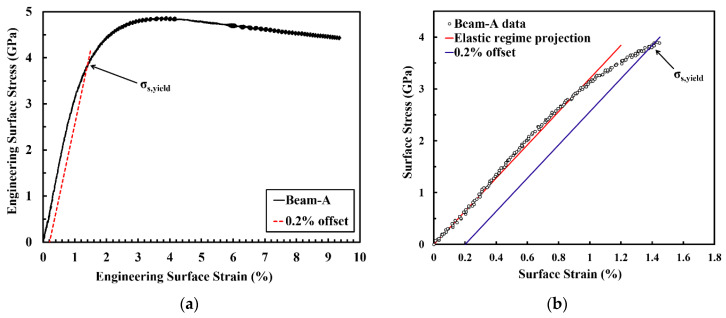
(**a**) Engineering surface stress-strain curve of micro beam-A (black solid line) and its 0.2% offset (red dashed line). (**b**) Detailed engineering surface stress-strain curve at the elastic-plastic transition regime equipped with an elastic regime projection (red line) and 0.2% offset (blue line).

**Figure 8 materials-13-02616-f008:**
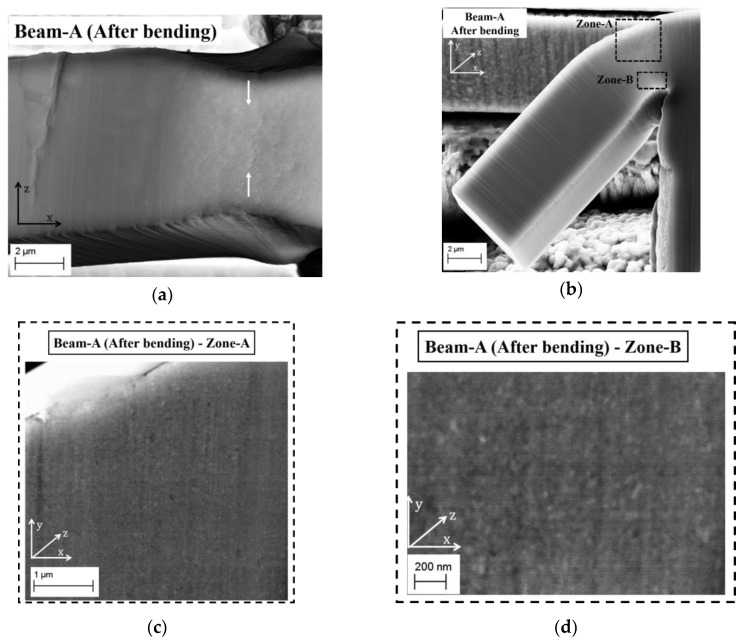
(**a**,**b**) Secondary electron images of micro beam-A after static bending test from (**a**) top and (**b**) side view of mechanical loading direction. (**c**,**d**) Back-scattered electron images of the microstructure of micro beam-A at two different positions, which are marked in (**b**).

**Figure 9 materials-13-02616-f009:**
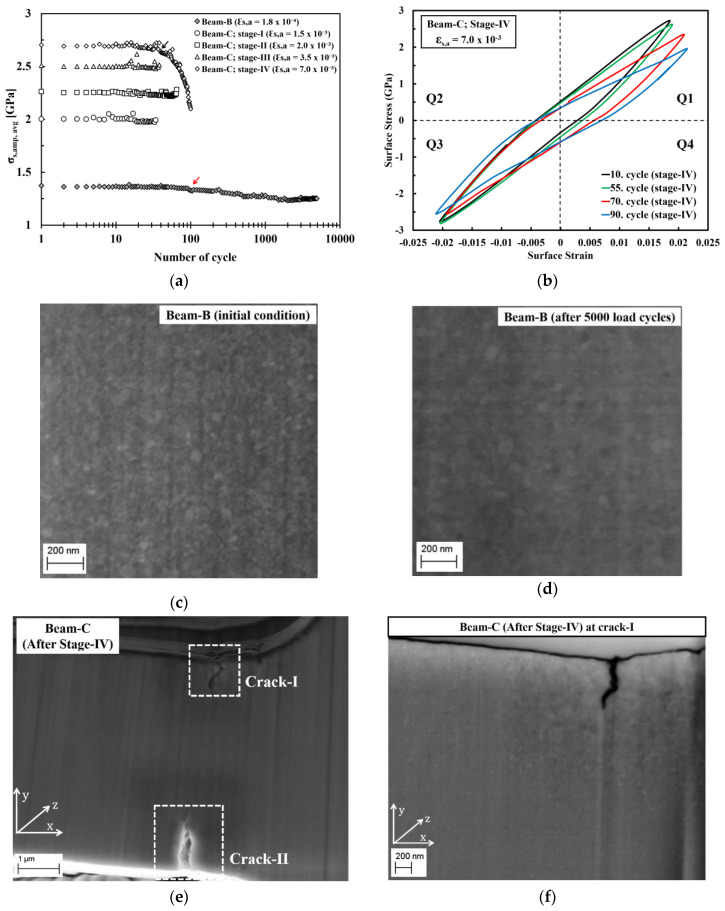
(**a**) Average surface-stress amplitude as a function of number of load cycle obtained from the cyclic micro-bending tests of micro beam-B and beam-C at different plastic surface strain amplitude. (**b**) Cyclic hysteresis loops of micro beam-C during cyclic bending test at stage-IV at the selected number of load cycles. (**c**,**d**) Back-scattered electron (BSE) images recorded at the same position on micro beam-B show microstructure images of (**c**) initial condition and (**d**) after 5000 load cycles without focused ion beam (FIB) polishing. (**e**) SE image shows a formation of cracks at two different position (crack-I and crack-II) on micro beam-C after cyclic tests of stage-IV. (**f**) BSE image shows a microstructure image of selected areas adjacent to the cracks on micro beam-C after FIB polishing at a position of crack-I.

**Table 1 materials-13-02616-t001:** Dimensions of micro bending beams with the length (L), width (W), thickness (T), and bending length (L_B_). Static micro bending test were carried out at the surface strain rate Є, while cyclic micro bending test were performed at multiple stage plastic strain amplitude ($\varepsilon_{s,a}$) and number of cycles (N).

Name of Sample	L (µm)	W (µm)	T (µm)	L_B_ (µm)	Static Loading	Cyclic Loading	
					Є (s^−1^)	$\varepsilon_{s,a}$	N (Cycles)
Beam-A	15.04	7.15	5.33	14.60	1.24 × 10^−2^	-	-
Beam-B	15.43	6.68	4.92	14.00	-	1.80 × 10^−4^	5000
Beam-C (4 stages)	15.77	6.75	5.26	13.92	-	1.50 × 10^−3^ (Stage-I)	35
						2.00 × 10^−3^ (Stage-II)	65
						3.50 × 10^−3^ (Stage-III)	40
						7.00 × 10^−3^ (Stage-IV)	100

**Table 2 materials-13-02616-t002:** Microhardness and average grain size of the annealed samples. Grain size measurements were determined based on the microstructure images from the scanning electron microscope (SEM) measurements.

Microhardness (GPa)				Grain Size (nm)		
	300 °C	450 °C	600 °C	300 °C	450 °C	600 °C
1 h	4.89 ± 0.06	4.43 ± 0.07	3.94 ± 0.06	<50 *	122.0 ± 19.4	159.6 ± 66.2
5 h	4.67 ± 0.18	4.40 ± 0.06	3.69 ± 0.06	<50 *	124.0 ± 40.5	183.6 ± 67.2
24 h	4.56 ± 0.06	4.31 ± 0.05	3.25 ± 0.12	<50 *	127.0 ± 39.0	218.6 ± 73.2

* Rough estimation from SEM images.

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
