# Peer review of "Microstructure Evolution and Mechanical Stability of Supersaturated Solid Solution Co-Rich Nanocrystalline Co-Cu Produced by Pulsed Electrodeposition"

_materials, 2020, doi:10.3390/ma13112616_

Round 1
Reviewer 1 Report
Useful data were obtained on the structural transformations of the Co-Cu alloy film in a wide temperature range.
Comments and suggestions for the article:
- It is necessary to describe in more detail the procedure for the synthesis of films.
- It is necessary to indicate in which atmosphere the heating was carried out (air, inert gas, vacuum?). How can the presence of oxygen during heating affect the results?
- It is necessary to explain the reasons for using the parameters of this film indicated in the article (28 at.% Cu, films of up to 400 μm) and discuss how the variation of these parameters can affect the results.
After making the specified clarifications, the article may be published.
Author Response
Point 1: It is necessary to describe in more detail the procedure for the synthesis of films.
Response 1: The details of the procedure for the synthesis of nanocrystalline Co-Cu films have been added into the manuscript at the line of 75-82:
“Nanocrystalline Co-Cu films up to 400 µm were deposited through the PED on Cu disk cathodes (surface diameter: 12 mm) through a Pt anode under specific parameters and conditions of deposition (current density: 1000 A/m2; pulse period: 2 ms; break period: 18 ms; temperature: 40 °C). Pulsed electrodeposition was performed in the electrolyte containing 112.20 g/L CoSO4.7H2O, 6.38 g/L CuSO4, 56.44 g/L C4H4KNaO6.4H2O, 142.04 g/L Na2SO4, 18.55 g/L H3BO3, 2.00 g/L C7H5NO3S (Saccharin), and 0.20 g/L C12H25NaO4S (sodium dodecyl sulfate). In the present work, the determination of parameters of deposition and composition of electrolytes are based on previous work by Müller [34] and continued with the preliminary experiment to obtain compact and homogeneous deposits.”
Point 2: It is necessary to indicate in which atmosphere the heating was carried out (air, inert gas, vacuum?). How can the presence of oxygen during heating affect the results?
Response 2: Isothermal annealing was carried out in the vacuum at a pressure of 10-6 mbar, thus, influence from surrounding gas such as oxygen, nitrogen, etc. is negligible. Moreover, SEM-EDS measurement confirmed that no formation of oxide on the samples. The changes have been added into the manuscript at the line of 83-84:
“The deposited samples were subjected to isothermal annealing in vacuum at a pressure of 10-6 mbar at various temperatures (300oC - 600oC) for different periods of time (1 h – 64 h) to investigate the thermal stability and structural transformation.”
Point 3: It is necessary to explain the reasons for using the parameters of this film indicated in the article (28 at.% Cu, films of up to 400 μm) and discuss how the variation of these parameters can affect the results.
Response 3: According to the master thesis by Timo Müller (2014) and preliminary research, the current density of 1000 A/m2 is the ‘safe’ number to produce a pure nanocrystalline structure with homogeneous chemical composition, microstructure, and low porosity of deposits. Deposition at 500 A/m2 shows a mixture of nano and ultrafine grain. On the other hand, deposition at 1500 A/m2 produces a pure nanostructure but it may cause a porous formation on the deposits due to high current density. One figure (Figure 1) and one paragraph (line 136-151) have been added in the manuscript to explain the variation of parameters on the deposited Co-Cu alloys. For information, the detailed effect of the parameter of deposition on the mechanical properties of nanocrystalline Co-Cu will be presented in other publications.
“Figure 1a depicts a cross-section image and EDS map of compact and porosity-free of nanocrystalline Co-Cu film deposited at a current density of 1000 A/m2 and pulse period 2 ms (duty cycle: 10%). EDS line profile in Figure 1b shows a quiet homogeneous concentration of Co and Cu along a cross-section of nanocrystalline Co-Cu (28 at.% Cu) film. EDS line profile confirms that the cobalt concentration drops up to ~15 at.% from area near Cu-substrate to the surface of the nanocrystalline film. Thus, deposition for more than 400 µm is not preferred due to the inhomogeneous concentration of Co and Cu. The deposition of nanocrystalline Co-Cu at higher Cu concentration is possible by reducing current density. However, a back-scattered electron image of Co-Cu (45 at.% Cu) alloy, deposited at a current density of 500 A/m2 and pulse period of 2 ms (duty cycle: 10%), shows inhomogeneous microstructure in which a mixture of the ultrafine-grained and nanocrystalline structure is observed (see Figure 1c). Deposition at a higher current density and duty cycle [for example at a current density of 1500 A/m2 and duty cycle 20% (see Figure 1d)], lead to the formation of microporous (mark with blue arrows in Figure 1d) and inhomogeneous chemical composition. Therefore, to achieve homogeneous microstructure and chemical composition, as well as compact and porosity-free deposits, all samples used in this paper were deposited at a current density of 1000 A/m2 and a pulse period of 2 ms (duty cycle: 10%).”

Reviewer 2 Report
The author have conducted a very interesting research that I think will be on interest to the readership. I believe the paper can be improved if the following changes are made;
- The authors indicated in the introduction that more research about microstructural changes at lower temperatures is required. And that information about mechanical properties of nanocrystalline Co-Cu is also limited. However greater than 60% of the literature used are older that 10 years. It is recommended that the authors expand the literature review and improve the currency of the references cited.
- Fig 1B was not discussed in the text before 1C and 1D
- Fig 3B was discussed before Fig 2
- The image quality for Fig 3 is very poor, this should be improved or replaced with a sharper image.
Author Response
Point 1: The authors indicated in the introduction that more research about microstructural changes at lower temperatures is required. And that information about mechanical properties of nanocrystalline Co-Cu is also limited. However greater than 60% of the literature used are older that 10 years. It is recommended that the authors expand the literature review and improve the currency of the references cited.
Response 1: Most of literature about mechanical properties and microstructural changes of nanocrystalline Co-Cu was taken from papers published on between 2014-2018 (References 4 – 11). Moreover, study on nanocrystalline Co-Cu is rarely found, thus the references are limited. Here are the references:
- Berkowitz, A.E.; Mitchell, J.R.; Carey, M.J.; Young, A.P.; Zhang, S.; Spada, F.E.; Parker, F.T.; Hutten, A.; Thomas, G. Giant magnetoresistance in heterogenous Cu-Co alloys. Physical Review Letters 1992, 68, No. 25, 3745-3748. [CrossRef]
- Bachmaier, A.; Krenn, H.; Knoll, P.; Aboulfadl, H.; Pippan, R. Tailoring the magnetic properties of nanocrystalline Co-Cu alloys prepared by high pressure torsion and isothermal annealing. Journal of Alloys and Compound 2017, 725, 744-749. [CrossRef]
- Bachmaier, A.; Motz, C. On the remarkable thermal stability of nanocrystalline cobalt via alloying, Materials Science & Engineering A 2014, 624, 41-51. [CrossRef]
- Bachmaier, A.; Pfaff, M.; Stolpe, M.; Aboulfadl, H.; Motz, C. Phase separation of supersaturated nanocrystalline Cu-Co alloy and its influence on thermal stability, Acta Materialia 2015, 96, 269-283. [CrossRef]
- Bachmaier, A.; Stolpe, M.; Müller, T.; Motz, C. Phase decomposition and nano structured evolution of metastable nanocrystalline Cu-Co solid solution during thermal treatment. IOP Conf. Series: Materials Science and Engineering 2015, 89, 012017, 1-8. [CrossRef]
- Bachmaier, A.; Rathmayr, G.; Schmauch, J.; Schell, N.; Stark, A.; De Jonge, N.; Pippan, R. High strength nanocrystalline Cu–Co alloys with high tensile ductility. Journal of Materials Research 2019,34(1), 58-68. [CrossRef]
- Nakamoto, Y.; Yuasa, M.; Chen, Y.; Kusuda, H.; Mabuchi, M. Mechanical properties of nanocrystalline Co-Cu alloy with high-density fine nanoscale lamellar structure. Scripta Materialia 2008, 58, 731-734. [CrossRef]
- Pratama, K.; Kunz, F.; Lindner, L.; Schmauch, J.; Mücklich, F.; Motz, C. Thermal stability, phase decomposition, and micro-fatigue properties of pulsed electrodeposited nanocrystalline Co-Cu. Procedia Structural Integrity 2019, 23, 366-371. [CrossRef]
Point 2: Fig 1B was not discussed in the text before 1C and 1D. Fig 3B was discussed before Fig 2
Response 2: Due to the addition of 1 Figure in the manuscript, major changes have been made according to this issue. The changes are:
Figure 1 is a new Figure added in the manuscript.
Figure 1 (previous manuscript) become Figure 2 (current manuscript)
Figure 2 (previous manuscript) become Figure 3 (current manuscript)
Figure 3 (previous manuscript) become Figure 4 (current manuscript)
Figure 4 (previous manuscript) become Figure 5 (current manuscript)
Figure 5 (previous manuscript) become Figure 6 (current manuscript)
Figure 6 (previous manuscript) become Figure 7 (current manuscript)
Figure 7 (previous manuscript) become Figure 8 (current manuscript)
Figure 8 (previous manuscript) become Figure 9 (current manuscript)
Changes have been made according to point 2:
Figure 1B (previous manuscript) was changed to Figure 2E.
Fig 1C (previous manuscript) was separated to Fig 2B (as deposited) and Fig 2F (annealed)
Fig 3B (previous manuscript) was changed to Fig 2D.
Point 3: The image quality for Fig 3 is very poor, this should be improved or replaced with a sharper image.
Response 3: In our opinion, the quality of the pictures is good enough. The picture was taken by the SEM-BSE detector at the magnification of 30,000x and the grain size is very small less (< 200 nm). It is difficult to produce sharper images. However, we tried to improve the picture quality, you could look at the new version of Fig 4 in the manuscript of page 9.

Reviewer 3 Report
May 20, 2020
Report on the paper #Materials-816923
Title “Microstructure Evolution and Mechanical Stability of Supersaturated Solid Solution Co-rich Nanocrystalline Co-Cu Produced by Pulsed Electrodeposition”
Authors: Pratama et al.
which was submitted for publication to Materials.
This manuscript describes a thorough study of the microstructure and the mechanical properties of Co-Cu electrodeposits both in the as-deposited and in the annealed states. The findings are important for the field and the study has been properly described in the manuscript. The English is also fine mostly. It is definitely suggested for publication in Materials.
However, the authors should be given the chance to perform some revision to improve the paper before final acceptance. Some comments are given below for the revision.
Specific questions and comments on the content:
- With reference to Figure 1, the authors write at the top of page 5 that annealing at 300 oC for 64 hours does not cause “significant microstructural changes” with respect to the as-deposited state according to the TEM and XRD results. On the other hand, we can see that the third ring of the SAED pattern in Fig. 1a (as-deposited state) is missing in the SAED pattern of Fig. 1b (annealed state) and the specified lattice parameters in the SAED patterns are also different for the two states. The XRD patterns of Fig. 1d also indicate a change since the position of the main observed Bragg reflection for the annealed state is shifted significantly with respect to the as-deposited state. Have you evaluated the line broadening if it also changes upon annealing? Please reconsider the evaluation of Fig. 1 in view of these remarks.
- For the evaluation of the single observed Bragg peak between the expected positions of the pure fcc-Cu and fcc-Co (111) peaks, please consult the paper by Michaelsen [Philos. Mag. A 72, 813 (1995)] which explains whether in such a case we have to do with a solid solution of Co and Cu or with a nanoscale mixture of Co and Cu grains. In assigning the Bragg peaks to individual Co and Cu phases, please take into account that the fcc-Co(111) peak position is practically the same as the hcp-Co(002) peak, please check this in the JCPDS cards. This information should be kept in mind when trying to establish if there is any hcp-Co phase in the annealed samples (Fig. 4). The considerations from the Michaelsen paper are important also for the evaluation of the XRD patterns of Fig. 4.
- Below Fig. 1, the authors write: “The formation of twins is believed due to effect of chemical additives (saccharine and sodium dodecyl sulfate) during the electrodeposition process as demonstrated in a previous paper [33].” However, Ref. 33 used thiourea as additive not the ones you listed, so please modify the ext. On the other hand, it might be relevant to cite here also Ref. 31 in which the influence of saccharin and formic acid on twin formation was investigated [and also in a subsequent paper by these latter authors: Kolonits et al., Surf. Coat. Technol. 349, 611 (2018)].
- The investigated electrodeposits were fairly thick (up to 400 /um thickness). Although pulsed electrodeposition may yield homogeneous films even along the thickness, at such high thicknesses there may be still some concerns about possible microstructure differences between the substrate and solution sides of the deposit. On the other hand, in the experimental section, there was no mention about whether TEM was carried out in planar mode or in cross-sectional mode. Please specify how the TEM sample thinnig was carried out and if there were any changes along the thickness if cross-sectional studies were made.
Comments on English
- Page 2, towards the end of the third paragraph: “The principle mechanical properties” -- > “The principal mechanical properties”.
- Page 2, last paragraph, second line: “films of up to 400 /um were deposited” -- > “films up to 400 /um thickness were deposited”; line 4: “Deposited samples were employed to isothermal annealing” -- > “The deposited samples were subjected to isothermal annealing”.
Technical comments:
- When you cite two neighboring references like [25-26], it should be rather written as [25,26].
- Caption to Fig. 1, beginning of line 2: “deposited” -- > “as-deposited”.
- It is exaggerated to specify the grain size to two digits after the comma: “22.71 +/- 8.16” -- > “22.7 +/- 8.2”
Author Response
Point 1: With reference to Figure 1, the authors write at the top of page 5 that annealing at 300oC for 64 hours does not cause “significant microstructural changes” with respect to the as-deposited state according to the TEM and XRD results. On the other hand, we can see that the third ring of the SAED pattern in Fig. 1a (as-deposited state) is missing in the SAED pattern of Fig. 1b (annealed state) and the specified lattice parameters in the SAED patterns are also different for the two states. The XRD patterns of Fig. 1d also indicate a change since the position of the main observed Bragg reflection for the annealed state is shifted significantly with respect to the as-deposited state. Have you evaluated the line broadening if it also changes upon annealing? Please reconsider the evaluation of Fig. 1 in view of these remarks.
Response 1: We evaluated both of diffraction from XRD and SAED measurement. We understand that there is a “lattice distortion” on between as-deposited and annealed samples in both of XRD and SAED, however we could not see any clear evidence of grain changes on the TEM bright field image. Thus, we called it “no-significant microstructure changes”. In order to make it more obvious, we changed the term “no significant microstructural changes” to “no significant grain changes”. We also believed that the XRD-peak distortion on sample annealed at 300C for 64 hours is not single-handedly caused by spinodal decomposition. Other factors such as recovery process and internal stress relaxation could contribute also to the XRD-peak distortion. Thus, we did APT measurement to confirm the chemical or spinodal decomposition. We added this information in the manuscript in line 187-192.
“The spinodal decomposition of the Co-Cu solid solution at the nanometer scale is expected due to the observation of a peak distortion of the {111} and {200} planes (see Figure 2c) on XRD and SAD pattern. However, it is also believed that the distortion is not single-handedly caused by spinodal decomposition. It could be contributed also from other factors or processes during isothermal annealing at 300°C for 64 h such as relaxation of grain boundaries, defects, and internal stresses.”
Point 2: For the evaluation of the single observed Bragg peak between the expected positions of the pure fcc-Cu and fcc-Co (111) peaks, please consult the paper by Michaelsen [Philos. Mag. A 72, 813 (1995)] which explains whether in such a case we have to do with a solid solution of Co and Cu or with a nanoscale mixture of Co and Cu grains. In assigning the Bragg peaks to individual Co and Cu phases, please take into account that the fcc-Co (111) peak position is practically the same as the hcp-Co (002) peak, please check this in the JCPDS cards. This information should be kept in mind when trying to establish if there is any hcp-Co phase in the annealed samples (Fig. 4). The considerations from the Michaelsen paper are important also for the evaluation of the XRD patterns of Fig. 4.
Response 2: We understand that according to paper by Michaelsen [Philos. Mag. A 72, 813 (1995)], the presence of single fcc-Cu or fcc-Co may contribute to the peak intensity of solid solution Co-Cu. However, according to APT measurement (Fig. 2A), we did not find any single phase of Co or Cu such as Co or Cu segregation/precipitate in nanoscale for less than 10 nm in the as-deposited state. Thus, here we could not find the presence of pure single fcc-Cu or fcc-Co. This type of Co or Cu chemical/spinodal decomposition is found on the APT measurement of the sample annealed at 300°C for 64 h (Fig. 2B). XRD measurement of the annealed sample still shows a single peak solid solution on between fcc-Co and fcc-Cu, but a peak position is shifted compared to the as-deposited state. APT measurement of an as-deposited state also reveals that the homogeneity of Co-Cu is not perfect in the nanoscale.
According to JCPDS cards, the position of fcc-Co (111) and hcp-Co (002) is not the same, thus we should be able to distinguish the individual peaks. Thus, our analysis is focussed on between 40° - 50° to get more details on this range [fcc-Co (111) at 44.22° and hcp-Co (002) at 44.76°]. In the manuscript, we discuss the presence of hcp-Co in the plane of (101) at 47.57° which can be clearly observed from form XRD pattern in Fig. 5. Thus, we added information in the manuscript as “hcp-Co {101}”.
Point 3: Below Fig. 1, the authors write: “The formation of twins is believed due to effect of chemical additives (saccharine and sodium dodecyl sulfate) during the electrodeposition process as demonstrated in a previous paper [33].” However, Ref. 33 used thiourea as additive not the ones you listed, so please modify the ext. On the other hand, it might be relevant to cite here also Ref. 31 in which the influence of saccharin and formic acid on twin formation was investigated [and also in a subsequent paper by these latter authors: Kolonits et al., Surf. Coat. Technol. 349, 611 (2018)].
Response 3: Thanks for the suggestion, we have made it also into the list of references [Ref 33]. However, Ref. 33 [Kolonits (2018)] discusses the effect of additives content on the deposition of nanocrystalline Ni by direct current. Thus, we kept the previous Ref. 33 [Kumar (2013)] and changed it to the Ref. 46 on the list of references for discussion in the formation of lattice defect during the pulsed electrodeposition process.
Point 4: The investigated electrodeposits were fairly thick (up to 400 /um thickness). Although pulsed electrodeposition may yield homogeneous films even along the thickness, at such high thicknesses there may be still some concerns about possible microstructure differences between the substrate and solution sides of the deposit. On the other hand, in the experimental section, there was no mention about whether TEM was carried out in planar mode or in cross-sectional mode. Please specify how the TEM sample thinning was carried out and if there were any changes along the thickness if cross-sectional studies were made.
Response 4: The deposits have a relatively homogeneous grain size and composition at 400 µm (we have the EDS line in the cross-section). The TEM sample was carried out at the planar mode (not cross-section mode). This information has been added to the manuscript in line of 97.
Point 5: Page 2, towards the end of the third paragraph: “The principle mechanical properties” -- > “The principal mechanical properties”.
Response 5: This information or change has been added to the manuscript at line of 70.
Point 6: Page 2, last paragraph, second line: “films of up to 400 /um were deposited” -- > “films up to 400 /um thickness were deposited”; line 4: “Deposited samples were employed to isothermal annealing” -- > “The deposited samples were subjected to isothermal annealing”.
Response 6: This information or change has been added into the manuscript at the line of 75-76 and 83-84.
Point 6: When you cite two neighboring references like [25-26], it should be rather written as [25,26]. Caption to Fig. 1, beginning of line 2: “deposited” -- > “as-deposited”. It is exaggerated to specify the grain size to two digits after the comma: “22.71 +/- 8.16” -- > “22.7 +/- 8.2”.
Response 6: This information or change has been added to the manuscript.

Reviewer 4 Report
This manuscript investigated the thermal and mechanical stability of nanocrystalline Co-Cu film produced by pulsed electrodeposition. The authors present reliable experimental results from SEM, BSE, XRD, TEM, APT, and miro bending tests. English writing is sufficient, and the scientific background is quite strong. Although this manuscript looks similar to the article which recently published by the authors, this manuscript is improved by additional data obtained by APT. I have no doubts that this manuscript would be interesting to potential readers of Materials. I have a few minor comments.
1) In the Abstract, the last three sentences describing mechanical stability are not clear. Please, revise them.
2) There are so many errors in abbreviation. The authors should revise them before the next submission.
3) Figure 3b is written on page 4 earlier than Figure 2. Also, I could not find any descriptions of Figure 3a.
Author Response
Point 1: In the Abstract, the last three sentences describing mechanical stability are not clear. Please, revise them.
Response 1: In the manuscript, it is written in the following sentences:
“Static micro-bending tests show that the nanocrystalline Co-Cu alloy exhibits a very high yield strength and ductile behavior with no crack formation. A static micro-bending test also reported that large plastic deformation is observed, but no microstructure change is detected. On the other hand, observation on the fatigue resistance of nanocrystalline Co-Cu shows that grain coarsening is observed after conducting the cyclic micro-bending test.”
I think this short description in the abstract is clear enough to explain the mechanical properties of nanocrystalline Co-Cu. Here is the complete explanation: In this research static and cyclic mechanical tests were conducted through the micro bending test. According to a static micro bending test, microbeam of nanocrystalline Co-Cu is plastically deformed with no crack formation. The observed yield strength is between 2.8–3.9 GPa. However, no microstructure change observed in the area where plastic deformation occurred. On the other hand, microstructure observation after cyclic micro-bending test reported microstructural change in which grain coarsening is observed.
Point 2: There are so many errors in abbreviation. The authors should revise them before the next submission
Response 2: Which abbreviation has an error? I usually describe the abbreviation in the early part. For example, I explain the scanning electron microscope (SEM) in the line of 70 in the manuscript. Thus, I used SEM words in the rest of the paper. For the abbreviation of the name of the journal in the reference, I have revised them.
Point 3: Figure 3b is written on page 4 earlier than Figure 2. Also, I could not find any descriptions of Figure 3a.
Response 3: Due to the addition of 1 Figure in the manuscript, major changes have been made according to this issue. The changes are:
Figure 1 is a new Figure added in the manuscript.
Figure 1 (previous manuscript) become Figure 2 (current manuscript)
Figure 2 (previous manuscript) become Figure 3 (current manuscript)
Figure 3 (previous manuscript) become Figure 4 (current manuscript)
Figure 4 (previous manuscript) become Figure 5 (current manuscript)
Figure 5 (previous manuscript) become Figure 6 (current manuscript)
Figure 6 (previous manuscript) become Figure 7 (current manuscript)
Figure 7 (previous manuscript) become Figure 8 (current manuscript)
Figure 8 (previous manuscript) become Figure 9 (current manuscript)
Changes have been made according to point 3:
Fig 3B (previous manuscript) was changed to Fig 2D.
Fig 3A (previous manuscript) was removed.

Round 2
Reviewer 3 Report
May 29, 2020
Report on the paper #Materials-816923-v2
Title “Microstructure Evolution and Mechanical Stability of Supersaturated Solid Solution Co-rich Nanocrystalline Co-Cu Produced by Pulsed Electrodeposition”
Authors: Pratama et al.
which was submitted for publication to Materials
The authors have poperly complied in the revised mansucript with almost all issues raised in my previous report and I accept the explanations given.
There is one single issue on which I would like to ask the authors to give some explanation. Namely, as noted in the previous report, there is a clear shift in the main Bragg position between the as-deposited and the annealed state (cf. Fig. 2c). In line 195-199, the authors speak about some “peak distortion” in connection with Fig. 2c. However, the meaning of the term “peak distortion” is not defined and not clear for the reader what is behind. Please explain “peak distortion”. In my view, this term cannot be taken anyway as an explanation for the peak shift during annealing. A few sentences should be given as explanation for the peak shift at this point.
Some technical corrections are also necessary:
- Line 98: “in planar mode/section” à “in planar mode”.
Apparently, the authors used automatic numbering of references and at several points the message “Error! Reference source not found.” can be seen in the text (see lines 120, 126, 183, 248 and 274, but it is better to check the whole text).
Author Response
Point 1: There is one single issue on which I would like to ask the authors to give some explanation. Namely, as noted in the previous report, there is a clear shift in the main Bragg position between the as-deposited and the annealed state (cf. Fig. 2c). In line 195-199, the authors speak about some “peak distortion” in connection with Fig. 2c. However, the meaning of the term “peak distortion” is not defined and not clear for the reader what is behind. Please explain “peak distortion”. In my view, this term cannot be taken anyway as an explanation for the peak shift during annealing. A few sentences should be given as explanation for the peak shift at this point.
Response to point 1: We changed the term "peak distortion" and "distortion" in the sentence becomes "peak shift" for the XRD pattern and "lattice shift" for the SAD pattern. Some additional sentences have been added also to make it more obvious. Here are the current sentences:
"In comparison with the as-deposited state, only minor changes are detected from the XRD measurement (Figure 2c). Comparing the SAD pattern of as deposited (Figure 2a) and annealed (Figure 2e) samples, lattice shifts of the {111} and {200} planes are observed and the spinodal decomposition of the Co-Cu solid solution at the nanometer scale during annealing at 300°C is expected to cause the shift. Consequently, peak shifts of the {111} and {200} planes are observed from the XRD pattern (see Figure 2c). However, it is also believed that the lattice shift is not single-handedly caused by spinodal decomposition, but could be caused also by other factors or processes during isothermal annealing at 300°C for 64 h such as relaxation of internal stresses and grain boundaries, reducing of defect density, etc. However, investigation on internal stress, grain boundaries relaxation, and defect density in nanocrystalline Co-Cu is not conducted in this paper, in which observation is focussed on the spinodal decomposition of nanocrystalline Co-Cu. Thus, APT measurements are conducted to investigate the possible structural and chemical decomposition at 300oC.
Point 2: Line 98: “in planar mode/section” à “in planar mode”.
Response to point 2: An error has been fixed.
Point 3: Apparently, the authors used the automatic numbering of references and at several points the message “Error! Reference source not found.” can be seen in the text (see lines 120, 126, 183, 248, and 274, but it is better to check the whole text).
Response to point 3: The cross-references have been checked and some errors have been fixed.
